# Preparation of Graphene/ITO Nanorod Metamaterial/U-Bent-Annealing Fiber Sensor and DNA Biomolecule Detection

**DOI:** 10.3390/nano9081154

**Published:** 2019-08-12

**Authors:** Wen Yang, Jing Yu, Xiangtai Xi, Yang Sun, Yiming Shen, Weiwei Yue, Chao Zhang, Shouzhen Jiang

**Affiliations:** 1Collaborative Innovation Center of Light Manipulations and Applications in Universities of Shandong, School of Physics and Electronics, Shandong Normal University, Jinan 250014, China; 2Institute of Materials and Clean Energy, Shandong Normal University, Jinan 250014, China; 3Shandong Key Laboratory of Medical Physics and Image Processing and Shandong Provincial Engineering and Technical Center of Light Manipulations, Shandong Provincial Key Laboratory of Optics and Photonic Device, Jinan 250014, China

**Keywords:** localized surface plasmon resonance biosensor, U-bent fiber, ITO metamaterials, high sensitivity, specificity deteection

## Abstract

In this paper, a graphene/ITO nanorod metamaterial/U-bent-annealing (Gr/ITO-NM/U-bent-A)-based U-bent optical fiber local surface plasmon resonance (LSPR) sensor is presented and demonstrated for DNA detection. The proposed sensor, compared with other conventional sensors, exhibits higher sensitivity, lower cost, as well as better biological affinity and oxidize resistance. Besides, it has a structure of an original Indium Tin Oxides (ITO) nanocolumn array coated with graphene, allowing the sensor to exert significant bulk plasmon resonance effect. Moreover, for its discontinuous structure, a larger specific surface area is created to accommodate more biomolecules, thus maximizing the biological properties. The fabricated sensors exhibit great performance (690.7 nm/RIU) in alcohol solution testing. Furthermore, it also exhibits an excellent linear response (*R*^2^ = 0.998) to the target DNA with respective concentrations from 0.1 to 100 nM suggesting the promising medical applications of such sensors.

## 1. Introduction

In recent years, local surface plasmon resonance (LSPR) sensors have attached much attention in environmental pollution, food safety, drug testing, and military [1,2,3,4,5] due to their excellent electrical and optical characteristics [6,7]. Thus far, numerous fiber structures have been studied to enhance the performance of the LSPR sensor (e.g., tapered fiber structures [8,9], partially uncoated fiber [10], side polished fiber [11], grating fiber structures [12], D-shaped fiber structures [13], as well as U-bent structures [14]). Among them, U-bent structures are considered one of the most promising LSPR sensors due to advantages including excellent evanescent field (big index difference) and simple detection (small tip volume).

So far, various materials have been widely used as the sensing surface of the U-bent LSPR sensor, exhibiting great performance in the detection of water, gas, and even DNA molecules [15,16,17,18]. In 2017, it was reported that the hybridized structure of graphene and silver nanoparticles was adopted as the sensing surface of the U-bent fiber, exhibiting excellent ability in glucose and alcohol detections (Zhang et. al.) [19]. In 2018, the single gold nanoparticle structure was deposited on U-bent fiber as the sensing surface, which could effectively detect lead ion in water (Boruah et. al.) [20]. Though these sensors exhibited great performance, numerous challenges exist (e.g., easy to oxidize, small span of absorption spectrum, and high-cost). Undoubtedly, LSPR sensors will be significantly optimized if these challenges can be resolved.

The Gr/ITO-NM/U-bent-A based on U-bent fiber, ITO nanocolumns metamaterial and graphene, with many merits (strong anti-oxidation, spectrum regulated widely, and low-cost), can achieve DNA-specific detection. ITO is an ordinary metal oxide with oxidation resistance [21,22,23]. Besides, graphene not only has additional signal enhancement, but also has biomodification for its good bioaffinity [24,25]. Thus, the hybrid structure is presented as one of the structures of hyperbolic super materials (HMMs) [26,27,28,29,30]. Compared with the film or nanoparticles structure, the plasma oscillation mode has developed from plane to three-dimensional structure (among nanocolumns). The bulk plasma oscillation mode can bring stronger LSPR effect. Moreover, the discontinuous structure creates a larger specific surface area to accommodate more biomolecules, maximizing the biologicality of the structure. Furthermore, ITO successfully overcomes the shortcoming of easy oxidation, poor biological affinity, and high-cost compared with conventional LSPR structure. Given all the advantages, the proposed LSPR sensors exhibit great performance in the detections of alcohol (690.7 nm/RIU) and DNA (limit of detection down to 0.1 nM), for its stronger collective response [31], high sensitivity, biological affinity, as well as larger specific surface area [32].

## 2. Experimental

### 2.1. Material

Quartz fiber was used (1.65 mm diameter). Chloroauric acid, sodium hydroxide, glucose, and sodium potassium tartrate solution were purchased from China Pharmaceutical Co., Ltd., Shanghai, China. 99.9% alcohol was provided by Lanyu Co., Ltd., Jinan, China. The 50% hydrazine hydrate solution (N_2_H_4_) was purchased from China Pharmaceutical Co., Ltd. Graphene oxide dispersion (diameter: 50–150 nm) at a concentration of 0.5 g·L^−1^ was provided from Xianfeng Nano Technology Co., Ltd. Poly(allylamine hydrochloride) (PAH), provided by China Pharmaceutical Co., Ltd., was used to make a solution (1 g·L^−1^). Indium oxide (99.9%) and tin oxide (99.9%) and graphite powder were provided by Lanyu Co., Ltd. DNA listed in Table 1 was purchased from Sangon Biotech Co., Ltd., Shanghai, Chnia. The 1-Pyrenebutanoic acid succinimidyl ester (PBASE), phosphate buffer saline (PBS), ethanolamine (EA), and N, N-dimethylformamide (DMF) were provided by Aladdin Co., Ltd., Shanghai, China.

### 2.2. Preparation of Gr/ITO NM/U-Bent-A

The process for the synthesis of the Gr/ITO-NM/U-bent-A sensor is illustrated in Figure 1. As shown in Figure 1, quartz fibers were cut into 30 cm segments, and the surface was cleaned with alcohol and deionized water (DI water). Subsequently, U-bent fibers (inner diameter: ~1.3 mm) were rapidly formed using an alcohol lamp that heated the U-bent zone fixed with a bracket. Au nanoparticles with a relatively uniform gap on the U-bent fibers were obtained by annealing the Au film in a double temperature zone plasma-enhanced chemical vapor deposition (DT-PECVD) furnace (Ar 40 sccm; temperature 500 °C). The Au film (3.1 nm) was formed by the mixed solution, including 2 mL gold chlorate solution (20 g·L^−1^), 3 mL sodium hydroxide solution (8 g·L^−1^) and 0.5 mL sodium tartrate solution (25 g·L^−1^), and 0.4 mL glucose solution (2.5 g·L^−1^). The uniform ITO nanocolumn arrays were grown on the fiber in the low temperature zone (500 °C) through the reaction of the mixture (0.004 g indium oxide and 0.036 g tin oxide mixed with graphite powder in the same quality) from high temperature zone (840 °C) with a special environment (Ar 160 sccm environment for 80 min). Furthermore, graphene was deposited on the surface of ITO nanocolumns by immersion in poly(allylamine hydrochloride) (PAH) solution (1 g·L^−1^) 1 h, then Go dispersion (0.2 g·L^−1^; diameter: 50–150 nm) 5 h, and finally hydrazine (N_2_H_4_, 50%) 1 h. Such a sensor was fabricated after being cleaned with deionized (DI) water. 

### 2.3. DNA Detection

The prepared sensor was immersed in 5 mL PBASE solution for 12 h, and then it underwent a preliminary specific modification. PBASE, acting as a bridge between the graphene and probe DNA, was introduced because its pyrene group could form π–π stacking with graphene, and its succinimide portion could conjugate with probe DNA modified by –NH_2_. Then, probe DNA, full completementary DNA (FC-DNA), and non completementary DNA (NC-DNA) were used to prepare the solution at different concentrations required by PBS buffer (shaken by hand for 30 min until uniform). Afterwards, the sensor modified by PBASE was added to the solution of 5 mL probe DNA (1 μM) for 4 h to ensure sufficient reaction. Next, the sensor was immersed in complementary DNA for 60 min to ensure that probe DNA and FC-DNA to be bound effectively. In the meantime, the performance of Gr/ITO-NM/U-bent-A/0.04/20 was tested with FC-DNA liquids at different concentrations prepared previously. The same process was repeat using NC-DNA replace FC-DNA to perform the hybridization for the examination of the specificity of such a sensor.

### 2.4. Experimental Setup

The experimental schematic diagram for measuring resonant wavelength drift of U-type metamaterial sensor is presented in Figure 2. The white light source (tungsten lamp, ocean optical HL-2000) with emission wavelength of 360 to 2000 nm was introduced into proposed sensor. The LSPR shift of the sensor was measured with the optical fiber spectrometer (PG2000, IDeaoptics Instruments, Shanghai, China). The surface morphology of all the sensors was observed under a scanning electron microscopy (SEM, Sigma 500, ZEISS, Oberkochen, Germany). The ITO was verified by the X-ray diffraction (XRD). Besides, the graphene deposited on the surface was examined under a transmission electron microscopy (TEM JEM-3200FS, Tokyo, Japan).

## 3. Results and Discussion

### 3.1. Exploration of Gr/ITO-NM/U-Bent Sensor

The performance of sensors fabricated at different positions with the same reagent mixture was studied, and the optimal conditions (20 cm) were found through the comparison of the sensitivity of those sensors. As shown in Figure 3a, U-bent zones have similar inner diameters (1.3 mm), which makes the sensors the same in the evanescent field; 1.3 mm U-bent optical fibers were taken for experiments following the reports [33]. Two milliliters gold chlorate solution (20 g·L^−1^), 3 mL sodium hydroxide solution (8 g·L^−1^), and 0.5 mL sodium tartrate solution (25 g·L^−1^) were dripped to the plastic tube and then shaken vigorously. Subsequently, a 3 nm gold film was deposited on the U-bent zone after 0.4 mL glucose solution (2.5 g·L^−1^) was poured into the mixed solution for 15 min. After the above steps, the Au film (3.1 nm) was prepared, as shown in Figure 3b. Chemical deposition, compared with physical deposition on both sides, could make the Au film more uniform because the fiber had a certain curvature rather than a plane. In such a way, the errors introduced by the uneven structure could be avoided.

The mixture (indium oxide, tin oxide and graphite powder) reacted within 80 min to form ITO at a high temperature of 840 °C, and then the fabricated ITO was transmitted to the U-bent fibers (at a low temperature of 500 °C) covered with Au film by Ar airflow (160 sccm). Besides, comparative experiments based on different locations (12, 16, 20, and 24 cm away from the reagent mixture center) were performed to explore the optimal position conditions to fabricate different sensors. Sensors were labeled as ITO-NM/U-bent/0.04/12, ITO-NM/U-bent/0.04/16, ITO-NM/U-bent/0.04/20, and ITO-NM/U-bent/0.04/24, respectively. Figure 3c–f shows the absorption spectra of different sensors by alcohol solutions at different concentrations (refractive index (RI) from 1.3330 to 1.3634, respectively). In Figure 3c, the wavelength shift remained almost unchanged and the absorption spectra were irregular. Accordingly, we assumed that this phenomenon was attributed to the damage of the sensor at 12 cm due to the high temperature. The SEM figures require further verification. Figure 3d–f shows the absorption spectra by alcohol solutions of ITO-NM/U-bent/0.04/16, ITO-NM/U-bent/0.04/20, and ITO-NM/U-bent/0.04/24 sensors at different concentrations, proving that all sensors performed well. Taking 475 nm as the origin, it is observed that the absorption peak of the above sensors shifts from 475 to 490 nm, as shown in Figure 3g. Such shift in the absorption spectrum may be caused by the variations of morphology to be further characterized in the following SEM figures. Figure 3h shows the histogram of absorption peak depths and wavelength shifting distances at different locations. The results show that the wavelength shift (the response of the sensor to the change of external RI) of the ITO-NM/U-BENT/0.04/20 sensor is most obvious, which indicates that the sensor grown under this condition has the best sensitivity. Thus, 20 cm from the reaction zone was taken as the condition to fabricate the high performance sensor.

The optimal conditions (0.04 g reagent mixture) were found by comparing the sensitivity of those sensors fabricated with different masses of regent mixture at 20 cm. To enhance the ability of the sensors, different amounts of reagents mixture were compared. Figure 4a–d shows the absorption spectra of sensors grown at 20 cm when reagents mixture of the indium tin oxide mixture was 0.01–0.1 g (the ratio of indium oxide to tin oxide is 1:9). The shift of the absorption peak from 450 to 600 nm may be caused by different masses of reagents mixture varied the parameter of ITO nanocolumns (length, diameter, etc.), as shown in Figure 4a–d. However, the performance of the sensor was reduced, as shown in Figure 4d. We assumed that the length of the ITO nanocolumn exceeded the range of the LSPR, hindering the plasmon oscillation and causing the sensor to malfunction. Furthermore, the ITO-NM/U-bent/0.04/20 sensor has the best features, as shown in Figure 4e. Thus, the optimal condition was determined as when the total mass of reagents was 0.04 g. According to the comparison of the ITO absorption spectra, it was demonstrated that ITO, instead of gold, exerted the LSPR effect (Figure 4f).

### 3.2. Characterization of Gr/ITO-NM/U-Bent Sensor

XRD and SEM were used to characterize the surface of the different sensors, and XRD mass spectrometry analysis of the material on the U-bent metamaterial sensor was conducted, as shown in Figure 5a. Besides, in Figure 5a, the ITO was verified by 400, 600, and 800 (ITO) peaks and 111 (Au) peaks [34]. Furthermore, Figure 5b–e gives the ITO frontal images of ITO-NM/U-bent/0.04/12, ITO-NM/U-bent/0.04/16, ITO-NM/U-bent/0.04/20, and ITO-NM/U-bent/0.04/24 sensors. The cracks in Figure 5b were primarily caused by excessive temperature, thus increasing the transmittance and changing the refractive index of quartz optical fibers. In the meantime, such cracks explain the reason for the sensor performance damage in Figure 3c. Some laws of different sensor morphological features can be concluded according to Figure 5c–e. Figure 5c shows that the nanorod structure on the surface of the sensor was an array structure consisting of gold balls on the stigma and ITO nanopillars. With increasing distance, the diameter of nanocolumn decreased, and the length of nanocolumn was reduced. The diversification of ITO nanocolumns (the diameter of nanocolumns varied from 20 to 14 nm; the length of nanocolumns from 600 to 350 nm), explaining the difference of peak absorption spectra with distance. Furthermore, the ITO nanocolumns of ITO-NM/U-bent/0.1/20 sensor were too long (~1.5 μm), limiting the plasma oscillation (Figure 5f), which explains the abnormal absorption spectrum curve in Figure 4d. However, nanocolumn arrays was obviously not uniform. Moreover, many sensors fabricated by growing with Au film U-bent fiber appeared blue shift. We speculate that the position of gold nanoparticles is random when directly catalyzed by gold film, which leads to uneven ITO morphology. We proposed a method to make ITO nanocolumn more uniform by a preannealing progress which will control the distribution of ITO growing position.

### 3.3. Exploration of Gr/ITO-NM/U-Bent-A Sensor

A more uniform structure was characterized using the proposed method and the performance was enhanced again by depositing graphene on the surface of ITO nanocolumns. Furthermore, the Gr/ITO-NM/U-bent/0.04/20 varied as Gr/ITO-NM/U-bent-A/0.04/20 after preannealing process. The Au particles (14 nm), following the classical Gauss, were fabricated after such an annealing process, as shown in Figure 6a. The ITO nanocolumn arrays are much more uniform after 60 min annealing than the prepared sensors without preannealing progress by comparing the right and lift figures in Figure 6b. Though it did not achieve the effect of nanotemplate, this low-cost method was highly cost-effective. Figure 6c shows SEM of Gr/ITO-NM/U-bent-A/0.04/20 after graphene deposition. The arrows in the Figure 6c represent a uniform graphene layer. Though some parts of the sensor were over-stacked, most of them were still very uniform. Figure 6c suggests that the process of depositing graphene damaged the ITO partial nanocolumn, whereas it did not affect the performance of the sensor and even have an improvement. Figure 6d–f shows ITO-NM/U-bent/0.04/20, ITO-NM/U-bent-A/0.04/20, and Gr/ITO-NM/U-bent-A/0.04/20 corresponding to the absorption spectrum of different refractive indices of alcohol, and the wavelength shift of these sensors, in order, was 11 nm, 15 nm, and 21 nm. We found that the performance of the sensors was constantly improving after the proposal methods. It explained that the annealing process did help to improve the sensitivity of the sensors due to the improvement of nanoarray structure. And the deposition of graphene on the sensors also improved its performance mainly due to the relatively large specific surface area of graphene which could absorb more molecules. TEM was performed to further characterize the structure of graphene ITO nanocolumns hybrid structure in Figure 6g, showing a thin and uniform graphene layer (~2.8 nm) deposited on ITO nanocolumns. Silicon dioxide wafer was used to grow the same structure because the fiber area was too small to bring errors when Raman mapping was being tested. The Raman spectra of 2910 cm^−1^ peak was collected on the area of 20 × 20 μm^2^ in Figure 6h, indicating that the relatively uniform and large-scale distribution of graphene [35]. Besides, the insert pattern shows Raman diagrams of graphene and graphene oxide, the characteristic peak of 2910 cm^−1^ confirmed that the deposited matter on ITO nanocolumns was graphene compared with the characteristic peaks of graphene oxide. Obviously, the resonance wavelength displayed a red shift with the rise in RI, and the response of the resonance wavelength was linear with RI. Moreover, the high determination coefficients (*R*^2^) of ITO-NM/U-bent/0.04/20, ITO-NM/U-bent-A/0.04/20, and Gr/ITO-NM/U-bent-A/0.04/20 were 0.968, 0.974, and 0.989, respectively, as shown in Figure 6i. After calculation, the corresponding refractive indexes (RI) were 361.8, 493.4, and 690.7 nm/riu. Thus, these parameters implied that the proposed method is feasible. Furthermore, Gr/ITO-NM/U-bent-A/0.04/20 sensor was used to identify DNA as the final solution for the experiment.

In addition to sensitivity, reproducibility is also a standard for characterizing sensor performance. Therefore, we have reprepared the Gr/ITO-NM/U-bent-A/0.04/20 sensor 7 times, and randomly selected seven sensors for the detection in alcohol solution with RI 1.3330. Figure 7a shows the distribution of Gr/ITO-NM/U-bent-A/0.04/20 sensor for the RI 1.3330 alcohol absorption spectrum. The impregnation and morphology of each absorption spectrum are similar and have great uniformity. At the same time, we tested the seven different sensors in alcohol solution (RI: 1.3330–1.3634), and they all showed excellent sensitivity. In Figure 7b, we can clearly find that the amount of red shift is similar, which shows that the sensor has good reproducibility.

### 3.4. Detection and Analysis of DNA Molecules

DNA was adopted to study the performance of such a sensor for biological monitoring. PBASE acting as a bridge between the graphene and probe DNA was introduced because its pyrene group can form π–π stacking with graphene, and its succinimide portion can be conjugated with probe DNA modified by –NH_2_ (Figure 8a,b) [36,37].To obtain the limit of detection of such a sensor to DNA, FC-DNA at different concentrations was used to hybridize with probe DNA by base complementary pairing, as shown in Figure 8c. To test the specificity of such a sensor, NC-DNA was used to hybridize withe probe DNA. The sensor should not be affected by the NC-DNA because NC-DNA cannot be attached in the surface of the sensor as shown in Figure 8d.

The specificity and sensitivity of such a sensor for the detection of DNA was studied. From Figure 9a, we can clearly observe that lowest point of absorption spectrum for sensor + PBASE + probe DNA is 475 nm and there is almost no shift after adding the NC-DNA, which is nearly invariable and can be explained by the fact that there is a non-bonding reaction between the probe DNA and NC-DNA strands. Therefore, the refractive index change around the sensing region is relatively weak due to they cannot hybridize. Such phenomenon suggests that the obtained sensor has great performance in biological specificity. When FC-DNA at different concentrations were added to the sensor which attached probe DNA, the absorption peak showed the absorption spectrum shifts to right. The shift of absorption spectrum can be explained by the electron-rich feature of DNA, and it shows an excellent linear relationship (*R*^2^ = 0.998), which proves that the obtained sensor has great linear detection capability (Figure 9b,c). It is noteworthy that it is necessary to clean excess impurities and excess DNA strands with PBS buffer after each test (FC-DNA and NC-DNA). In this way, we can ensure the accuracy of the results. To describe the detail of the absorption spectrum shift, we made a movement amount histogram of FC-DNA and NC-DNA at different concentrations in Figure 9d. 

To further assess the performance of the sensor, the saturation and dynamics to biomolecules on metamaterial sensors were preliminarily explored. The responses of such sensors to FC-DNA with different concentration (1–100 nM) were detected and organized in real time. As shown in Figure 10a, the variations of absorption spectrum to 100 nM FC-DNA was taken as an example, which was shown in Figure 10b (the black curve) that presented the wavelength shift as a function of hybridization time. When the target DNA molecules were continuously captured by modified probe DNA, the RI around the sensing surface would increase [38]. Obviously, the sensor had an intense response to FC-DNA hybridization before 50 min, and then the response was gradually stabilized. This indicated that the DNA hybridization reaction gradually progressed to saturation after a vigorous and rapid reaction. Besides, the red shift of 0.1, 1, 10, and 100 nM FC-DNA were 6, 13, 18 and 25 nm, respectively. However, 1 μM NC-DNA did not vary over time, suggesting that its hybridization was weaker than that of FC-DNA. This reveals the saturation state that this sensor can achieve. In this study, insufficient effort was made to ensure the accuracy of the data to 70 min.

## 4. Conclusions

In the present study, a proposed LSPR sensor based on Gr/ITO-NM/U-bent-A was employed to detect DNA. The Au film was annealed to Au nanoparticles, and ITO nanocolumns were grown by double temperature zone DT-PECVD as catalyzed by Au nanoparticles. The proposal sensor was fabricated after depositing graphene using a chemical method, and it exhibited great performance (690.7 nm/RIU sensitivity) by testing alcohol solutions at different concentrations. For DNA detection, the significant red shift and linear relationship of DNA at different concentrations also suggested that the proposed LSPR sensor had an excellent ability to detect DNA. The results implied that such a LSPR sensor can be a promising candidate to biological detection, and serves as a feasible platform for biological identification due to its low-cost, high sensitivity, biological affinity, as well as strong oxidation resistance.

## Figures and Tables

**Figure 1 nanomaterials-09-01154-f001:**
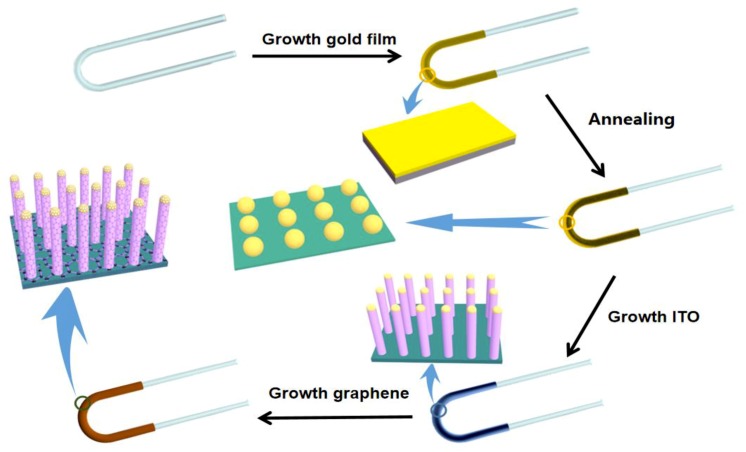
The schematic representation of the preparation procedure of Gr/ITO-NM/U-bent-A sensor.

**Figure 2 nanomaterials-09-01154-f002:**
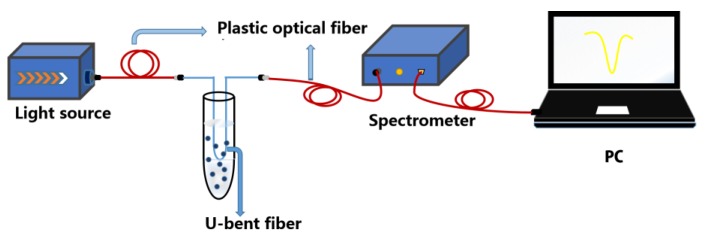
The schematic diagram for the experimental set-up used in the U-bent optical fiber LSPR sensor.

**Figure 3 nanomaterials-09-01154-f003:**
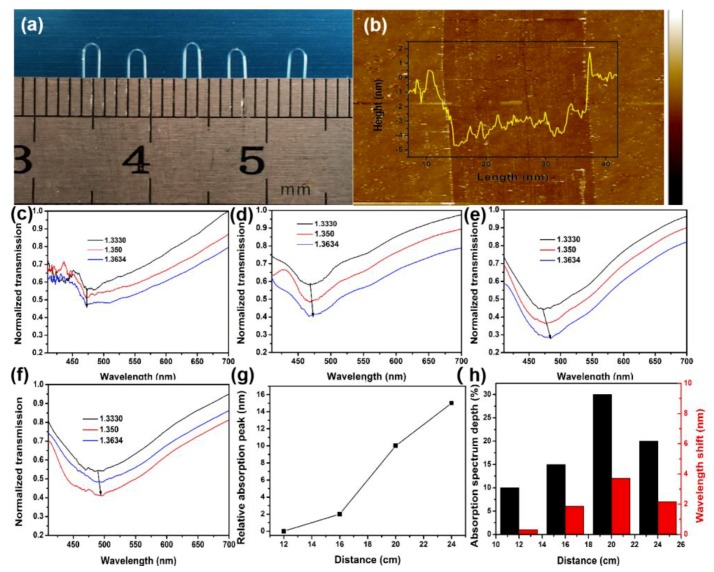
(**a**) The inner diameter of the U-bent bare optical fiber. (**b**) Atomic force microscopy (AFM) of Au film on the U-bent fiber. (**c**–**f**) The absorption spectra of ITO-NM/U-bent/0.04/12, ITO-NM/U-bent/0.04/16, ITO-NM/U-bent/0.04/20, and ITO-NM/U-bent/0.04/24 sensors in alcohol solution with RI from 1.3330 to 1.3634. (**g**) Absorption spectrum movement trend graph (475 nm as the origin). (**h**) Histogram of the absorption spectra depth and displacement distance corresponding to the different metamaterial sensors at different locations.

**Figure 4 nanomaterials-09-01154-f004:**
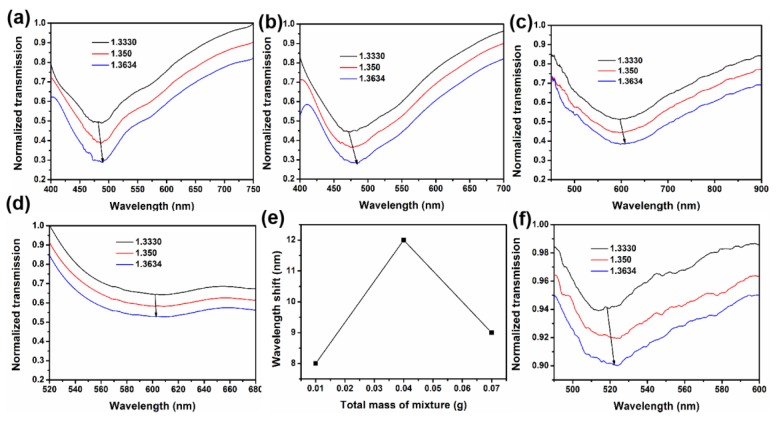
(**a**–**d**) Absorption spectra of ITO-NM/U-bent/0.01/20, ITO-NM/U-bent/0.04/20, ITO-NM/U-bent/0.07/20, and ITO-NM/U-bent/0.1/20 sensors in alcohol solution with RI from 1.3330 to 1.3634, respectively. (**e**) The broken line diagram of the shift of sensor at 20 cm with different growth conditions (reagents mixture: 0.01–0.07 g). (**f**) Absorption spectrum of U-bent with 3.1 nm Au film.

**Figure 5 nanomaterials-09-01154-f005:**
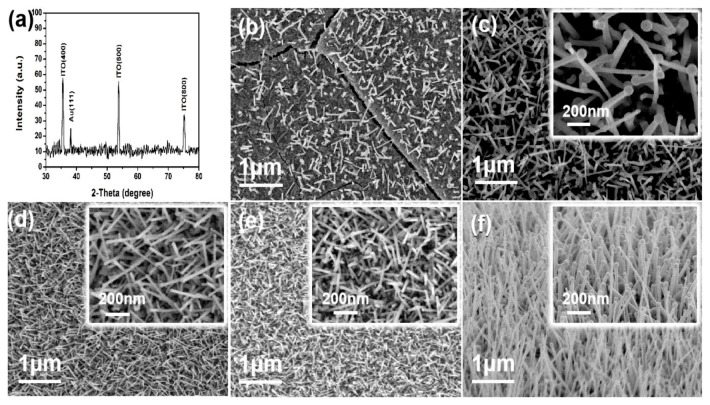
(**a**) XRD of ITO on the U-bent fiber. (**b**–**e**) ITO frontal images of ITO-NM/U-bent/0.04/12, ITO-NM/U-bent/0.04/16, ITO-NM/U-bent/0.04/20, and ITO-NM/U-bent/0.04/24. (**f**) The ITO frontal morphology of ITO-NM/U-bent/0.1/20.

**Figure 6 nanomaterials-09-01154-f006:**
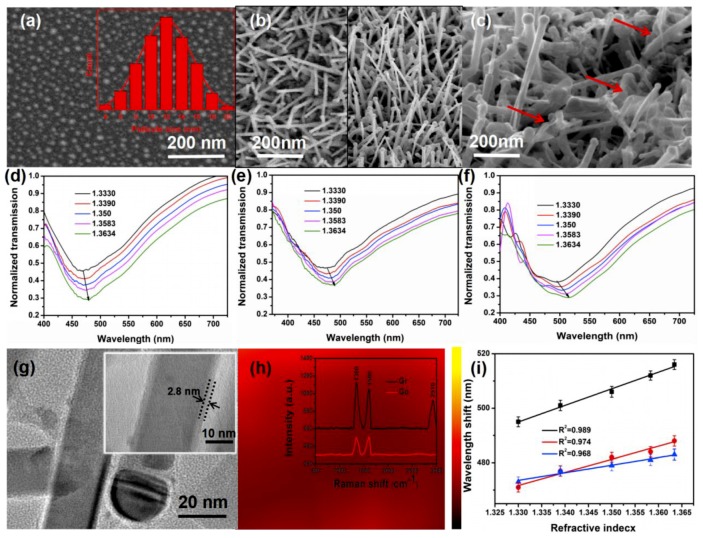
(**a**) Distribution of annealed Au particles. (**b**) Frontal profile of ITO (right figure shows the ITO with the preannealing process and the left exactly the opposite). (**c**) Surface morphology of Gr/ITO-NM/U-bent-A/0.04/20. (**d**–**f**) The absorption spectra of ITO-NM/U-bent/0.04/20, ITO-NM/U-bent-A/0.04/20, Gr/ITO-NM/U-bent-A/0.04/20 sensors in alcohol solution with RI from 1.3330 to 1.3634. (**g**) TEM diagram of Gr/ITO-NM/U-bent-A/0.04/20. (**h**) 2910 cm^−1^ peak mapping (20 × 20 μm^2^) of graphene and the insert pattern is Raman diagrams of graphene and graphene oxide on Gr/ITO-NM/U-bent-A/0.04/20 sensor. Appendix A shows 15 groups of Raman spectra collected from different sensors. (**i**) The relationship between resonance wavelength and refractive index of ITO-NM/U-bent/0.04/20, ITO-NM/U-bent-A/0.04/20, and Gr/ITO-NM/U-bent-A/0.04/20.

**Figure 7 nanomaterials-09-01154-f007:**
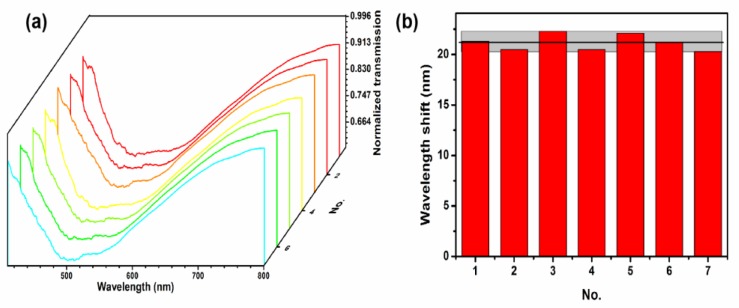
(**a**) The absorption spectra of Gr/ITO-NM/U-bent-A/0.04/20 from 7 different sensors in alcohol solution with RI 1.3330. (**b**) Wavelength shift histogram of different sensors.

**Figure 8 nanomaterials-09-01154-f008:**
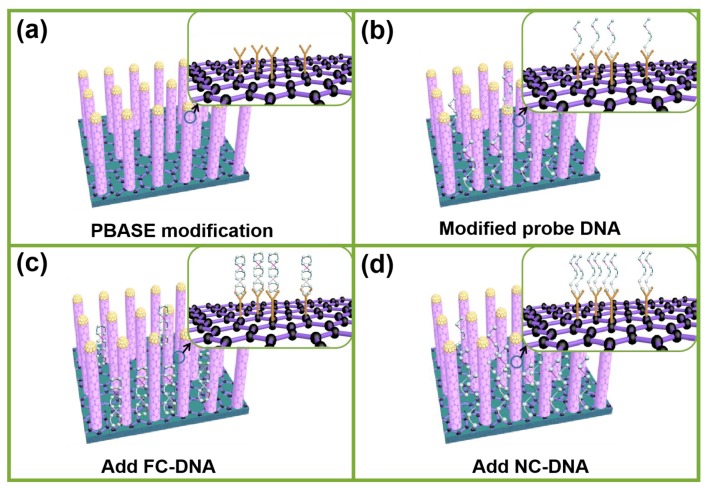
(**a**) The schematic diagram of PBASE modified on the metamaterial surface. (**b**) Schematic diagram shows the process of modifying probe DNA. (**c**) Schematic diagram after adding complementary DNA to the super material LSPR sensor. (**d**) Schematic diagram of adding unmatched DNA to the metamaterial LSPR sensor, with the enlarged details in the upper right corner.

**Figure 9 nanomaterials-09-01154-f009:**
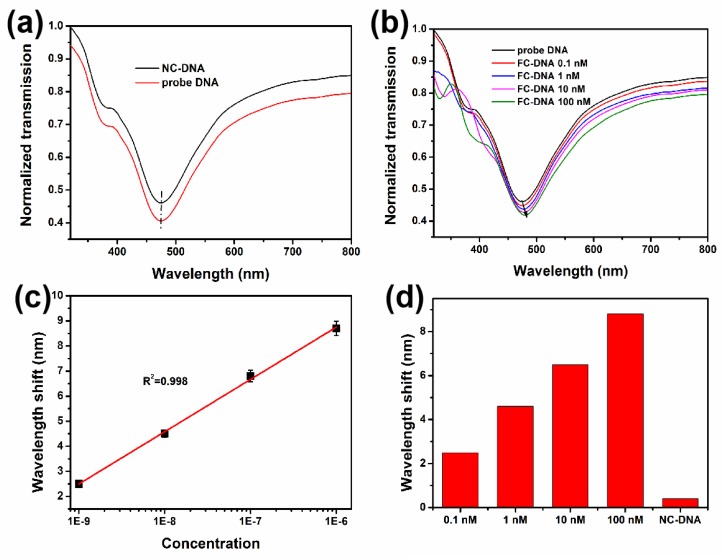
(**a**) LSPR spectrum of the sensor with PBASE after adding NC-DNA and probe DNA. (**b**) LSPR spectra of probe DNA and different concentrations of FC-DNA. (**c**) Resonance wavelength shift as a function of FC-DNA concentration. (**d**) Wavelength shift as a function of FC-DNA and NC-DNA, which is relative to probe DNA.

**Figure 10 nanomaterials-09-01154-f010:**
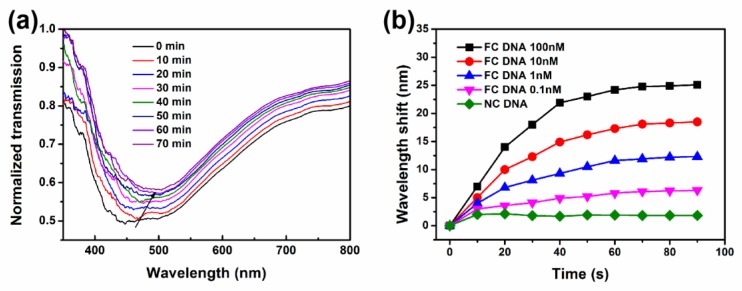
(**a**) The transmission spectra of 100 nM double-stranded DNA hybridization. (**b**) Wavelength drift of NC-DNA and different concentration of FC-DNA (0.1–100 nM) over time.

**Table 1 nanomaterials-09-01154-t001:** All the DNA used in this work.

Type	Sequences (24mer)
Probe DNA	5′-CTT CTG TCT TGA TGT TTG TCA AAC-3′
FC DNA	5′-GTT TGA CAA ACA TCA AGA CAG AAG-3′
NC DNA	5′-CAA CAT TCC GTT AAC CAT TCC CCA-3′

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
