# Peer review of "Preparation of Graphene/ITO Nanorod Metamaterial/U-Bent-Annealing Fiber Sensor and DNA Biomolecule Detection"

_nanomaterials, 2019, doi:10.3390/nano9081154_

Round 1

Reviewer 1 Report

In this paper, authors report localized surface plasmon resonance sensing based on U bent (U-bent) optical fibers. Despite significant data, the analysis and the overall performances fall far short of what may be typically expected. In this sense, significant changes need to be made before considering publication.

1.  A large part of the concept has been reported in previous publications of the same group. Please clarify the novelty of this work.

2. The schematic illustration provided in Figures 1 and 7 is misleading. The actual nanostructures that one can see in SEM images of Figures 5 and 6 are random in terms of many qualities including location, direction, and length. This would affect optical properties significantly in a way quite different from what the periodic structures in the schematic illustration would represent.

3. Graphene is known to be difficult to perform binding experiments in a predictable way. Authors need to address the reproducibility as well as the regeneration of the proposed concept. Statistical data need to be provided.

4. The LSPR sensing was evaluated with alcohol and DNA hybridization. I find the analysis extremely shallow and weak. In addition, the details of the tests are not revealed, for example there are many kinds of alcohol, the nature of which is not clearly discussed.

5. Usually, ambient detection such as using alcohol tends to amplify the detection signature. In this context, the performance results reported as 690 nm/RIU is not particularly high. Neither is the data of DNA hybridization deemed properly analyzed. Authors need to discuss this in comparison with other studies (even outside fiberoptics based SPR). There are hundreds of related works, if not thousands.

-       Fu, Hongyan, et al. "Graphene enhances the sensitivity of fiber-optic surface plasmon resonance biosensor." IEEE Sensors Journal 15.10 (2015): 5478-5482.

-       Ryu, Yeonsoo, et al. "Effect of coupled graphene oxide on the sensitivity of surface plasmon resonance detection." Applied optics 53.7 (2014): 1419-1426.

-       Grigorenko, A. N., Marco Polini, and K. S. Novoselov. "Graphene plasmonics." Nature photonics 6.11 (2012): 749.

-       Chung, Kyungwha, et al. "Systematic study on the sensitivity enhancement in graphene plasmonic sensors based on layer-by-layer self-assembled graphene oxide multilayers and their reduced analogues." ACS applied materials & interfaces 7.1 (2015): 144-151.

-       Wu, L., et al. "Highly sensitive graphene biosensors based on surface plasmon resonance." Optics express 18.14 (2010): 14395-14400.

6. For proper comparison of the data x and y axis should be identical in Figures 3, 4, and 6. Figures need to redone for improved understanding. For example, the overlap of an AFM image and height profile in Figure 3(b) is difficult to understand.

7. Correct grammatical errors and typos. For example, drop ‘And’ from the beginning of sentences. It is ‘localized surface plasmon resonance’. Check Ref. 25.

Reviewer 2 Report

The authors presented an interesting application of fiber sensor to DNA detection. For the benefit of riders, however, a number of points need clarifying and certain statement require for improving the quality of manuscript.

There is no commentary of figure 1 in the text. Author should explain it. 

Author should explain in detail the concentration of alcohol in the experiment. What is the unit of 1.3330, 1.350 and 1.3634 in the figures? Is it weight percent or volume percent? And why does the 1.350 in significant digits after the decimal point different from 1.3330 and 1.3634?

What is the solution that prepared FC-DNA and NC-DNA? The readers can not reproduce the experiment without information of the solution.

Biological sample contains a lot of extras things such as protein. Are there processes to reduce the nonspecific adsorption? Is there data on nonspecific adsorption? 

How is the repeatability of these results? Author should write about the repeatability.

In Table 1, the sequence should be represented by one line.

Line 118, line 269 and line 168, author should check the position of the period. Other places should be checked again.

The reported sensor is unique and interesting, however, the data is unreliable. There is no description about reproducibility and nonspecific adsorption. If there are these experimental data, I can recommend for publication.

Round 2

Reviewer 1 Report

Other than the second-to-last comment, I do not believe the revised manuscript to have properly addressed the comments (1 through 5).

Author Response

(Please see the attachment.)

Dear editor and reviewers:

Manuscript ID: 552604

We would like to thank you for giving us a chance to revise the paper, and also thank the reviewers for giving us constructive suggestions. These comments are of great significance to the improvement and improvement of our articles. Therefore, we have made a careful analysis of the revision opinions and actively cooperated with the revision. The corresponding revises have been marked with underline. The responses were listed as follow:

Reviewers' comments:

Reviewer Round1#1:Comments:In this paper, authors report localized surface plasmon resonance sensing based on U bent (U-bent) optical fibers. Despite significant data, the analysis and the overall performances fall far short of what may be typically expected. In this sense, significant changes need to be made before considering publication.

Reviewer Round2#1:Other than the second-to-last comment, I do not believe the revised manuscript to have properly addressed the comments (1 through 5).

Comment 1: A large part of the concept has been reported in previous publications of the same group. Please clarify the novelty of this work.

Response:Thank you for your valuable suggestion. It is very important for readers to understand the article.

1.ITO as a common metal oxide has been introduced in many articles, but ITO used in the field of SPR is not many and is basically based on film or particle structure. We use thermal carbon reduction method to grow ITO into nano-column arrays and use it for LSPR effect. Although there are other similar structures such as gold nanorods for SPR effect, we have not found relevant articles on ITO nanorod SPR sensors. Following is the SPR articles we found related to ITO materials, please review them.

(1) Dash, Jitendra Narayan, and Rajan Jha. "SPR biosensor based on polymer PCF coated with conducting metal oxide." IEEE Photonics Technology Letters 26.6 (2014): 595-598.

(2) Mishra, Satyendra K., Deepa Kumari, and Banshi D. Gupta. "Surface plasmon resonance based fiber optic ammonia gas sensor using ITO and polyaniline." Sensors and Actuators B: Chemical 171 (2012): 976-983.

(3) Patnaik, Amrit, K. Senthilnathan, and Rajan Jha. "Graphene-based conducting metal oxide coated D-shaped optical fiber SPR sensor." IEEE Photonics Technology Letters 27.23 (2015): 2437-2440.

(4) Patnaik, Amrit, K. Senthilnathan, and Rajan Jha. "Graphene-based conducting metal oxide coated D-shaped optical fiber SPR sensor." IEEE Photonics Technology Letters 27.23 (2015): 2437-2440.

2.    Hyperbolic metamaterial is a kind of synthetic material with special structure, which is widely accepted as multi-layer structure and nano-array structure. It has many advantages that ordinary membrane structure does not have. For example, discontinuous structure enlarges the specific surface area, and can absorb more special substances, especially biological molecules. Therefore, supermaterial The sensor of material structure can maximize the biology. Moreover, the supermaterial structure can bring about strong bulk plasma effect. The relevant literature is as follows.

(1) Ferrari, Lorenzo, et al. "Hyperbolic metamaterials and their applications." Progress in Quantum Electronics 40 (2015): 1-40.

(2) Lu, Dylan, et al. "Enhancing spontaneous emission rates of molecules using nanopatterned multilayer hyperbolic metamaterials." Nature nanotechnology 9.1 (2014): 48.

(3) Kabashin, A. V., et al. "Plasmonic nanorod metamaterials for biosensing." Nature materials 8.11 (2009): 867.

(4) Sreekanth, K. V., A. De Luca, and G. Strangi. "Negative refraction in graphene-based hyperbolic metamaterials." Applied Physics Letters 103.2 (2013): 023107.

3.    Compared with other structures such as D-type, taper, prism and so on, the formation of evanescent field of optical fiber U-type structure is due to the optical loss effect caused by different refractive index inside and outside. The main advantages are convenient detection, low cost and real-time monitoring. The relevant literature is shown below.

(1) Zhang, Chao, et al. "U-bent fiber optic SPR sensor based on graphene/AgNPs." Sensors and Actuators B: Chemical 251 (2017): 127-133.

(2) Sai, V. V. R., Tapanendu Kundu, and Soumyo Mukherji. "Novel U-bent fiber optic probe for localized surface plasmon resonance based biosensor." Biosensors and Bioelectronics 24.9 (2009): 2804-2809.

(3) Saikia, Rajib, et al. "Fiber‐Optic pH Sensor Based on SPR of Silver Nanostructured Film." AIP Conference Proceedings. Vol. 1147. No. 1. AIP, 2009.

(4) Liang, Gaoling, et al. "Plasma enhanced label-free immunoassay for alpha-fetoprotein based on a U-bend fiber-optic LSPR biosensor." Rsc Advances 5.31 (2015): 23990-23998.

Comment 2: The schematic illustration provided in Figures 1 and 7 is misleading. The actual nanostructures that one can see in SEM images of Figures 5 and 6 are random in terms of many qualities including location, direction, and length. This would affect optical properties significantly in a way quite different from what the periodic structures in the schematic illustration would represent.

Response: Thank you for your careful and attentive comments. It is a good advice to give a further description for the structure of the sensors, which can give a great guidance for the researchers worked in this field and can make readers know the process of experiment well.

1.    Figures 1 and 7 are just schematic diagram of this structure. The schematic diagram is a structure that we want to accomplish. It should be normal that there are some differences between the schematic diagram and the SEM diagram of the experimental results, because it is an ideal state. And one of the main directions we will introduce in this paper is that ITO nanoarrays have been improved to some extent with the improvement of the method. Although we have not achieved the perfect schematic structure, we have made progress. Fig. 6 (b) shows the improved structure. Compared with the previous arbitrary growth, we annealed the gold film first. The pre-annealing process can fix the growth point of ITO. In this way, the growth of ITO nanoarrays is not random. Because Figure 6 (a) shows that the particle distribution follows the Gauss distribution.

2.By fixing the thickness of gold, the dosage of reagent and the location of growth were explored. And we find the optimum conditions. We can learn from the article that the length and thickness of nano-column can be basically repeated with different reagents and growing positions. So the only uncertainty is the periodicity of the growth of the nano-column, which is improved by annealing the gold film. We found that after the thickness of gold, reagent and location are the same, the annealed sensor and the non-returned sensor are at the absorption peak.

Comment 3: Graphene is known to be difficult to perform binding experiments in a predictable way. Authors need to address the reproducibility as well as the regeneration of the proposed concept. Statistical data need to be provided.

Response: Thank you for your valuable comments and advice! Graphene as a very wide range of two-dimensional materials has many advantages. The chemical synthesis of graphene has been described in many articles, including its reproducibility, such as the following. In addition, we characterize the homogeneous thickness of graphene in Fig. 6 (g), (h), and we also do the renewability data of graphene. We did a lot of experiments these days, and we collected 15 groups data from different sensors. We processed the data and decided to add it to supplementary materials. This shows that the graphene grown by this method has good regeneration.

(1) Zhou, Weiwei, et al. "A general strategy toward graphene@ metal oxide core–shell nanostructures for high-performance lithium storage." Energy & Environmental Science 4.12 (2011): 4954-4961.

(2) Luo, Yongsong, et al. "Controlled synthesis of hierarchical graphene-wrapped TiO2@ Co3O4 coaxial nanobelt arrays for high-performance lithium storage." Journal of Materials Chemistry A 1.2 (2013): 273-281.

(3) Kong, Dezhi, et al. "Scalable synthesis of graphene-wrapped Li4Ti5O12 dandelion-like microspheres for lithium-ion batteries with excellent rate capability and long-cycle life." Journal of Materials Chemistry A 2.47 (2014): 20221-20230.

2.In this paper, the biological properties of graphene is one of the main advantages, which makes it possible for sensors to successfully detect biomolecules. The mechanism of graphene and biological DNA molecular detection are described in lots of published articles: PBASE that worked as a bridge between the graphene and probe DNA was introduced for its pyrene group can formed π-π stacking with graphene and its succinimide portion can conjugated with probe DNA modified by -NH2. The graphene layer prepared by chemical methods also has good biological effects and reproducibility, which is also reflected in our previous papers. And many excellent articles are about binding biomolecules with graphene, and the repeatability of this binding is also explained. The relevant articles are as follows, and we have cited several related articles in the article.

(1) Cai, Bingjie, et al. "Ultrasensitive label-free detection of PNA–DNA hybridization by reduced graphene oxide field-effect transistor biosensor." ACS nano 8.3 (2014): 2632-2638.

(2) Xu, Shicai, et al. "Real-time reliable determination of binding kinetics of DNA hybridization using a multi-channel graphene biosensor." Nature communications 8 (2017): 14902.

(3) Sun, Yang, et al. "Suspended CNT-Based FET sensor for ultrasensitive and label-free detection of DNA hybridization." Biosensors and Bioelectronics 137 (2019): 255-262.

(4) Mei, Junchi, et al. "Molybdenum disulfide field-effect transistor biosensor for ultrasensitive detection of DNA by employing morpholino as probe." Biosensors and Bioelectronics 110 (2018): 71-77.

(5)Yang, Wen, et al. "Graphene-Ag nanoparticles-cicada wings hybrid system for obvious SERS performance and DNA molecular detection." Optics express 27.3 (2019): 3000-3013.

The corresponding modification has been made:

add the this figure to supplementary materials

Figure S6(h) shows 15 groups raman spectra collected from different sensors.

Graphene is a very common two-dimensional material. In order to ensure the accuracy of the experimental results, we have explored the reproducibility of chemical methods. Figure S6(h) shows 15 groups raman spectra of graphene. In the figure, the peaks of 1380, 1580 and 2910 cm-1 peaks indicate that we have prepared graphene, The height of each peaks only have small float. This uniform phenomenon indicates that the chemical method for preparing graphene has good reproducibility.

Comment 4: The LSPR sensing was evaluated with alcohol and DNA hybridization. I find the analysis extremely shallow and weak. In addition, the details of the tests are not revealed, for example there are many kinds of alcohol, the nature of which is not clearly discussed.

Response:Thank you for your valuable comments and advice! Your valuable suggestions are very important to improve the overall level of the article, so we value your valuable suggestions and make many improvements. Thanks again.

The corresponding modification has been made:

1.Addition of LSPR Effect (Alcohol , DNA and LSPR) Principle

1. It can ensure great sensitivity because intense collective response originates from the uniform nanocolumns array. Moreover, the discontinuous structure creates a larger specific surface area to accommodate more biomolecules, maximizing the biologicality of the structure.

Compared with film or nanoparticles structure, the plasma oscillation mode has developed from plane to three-dimensional structure (among nanocolumns). This bulk plasma oscillation mode can bring stronger LSPR effect. Moreover, the discontinuous structure creates a larger specific surface area to accommodate more biomolecules, maximizing the biologicality of the structure.

2. The results suggest that ITO-NM/U-BENT/0.04/20 sensor had the optimal performance. Thus, 20 cm from the reaction zone as a condition to fabricate the high performance sensor.

The results show that the wavelength shift (the response of the sensor to the change of external RI) of ITO-NM/U-BENT/0.04/20 sensor is most obvious, which indicates that the sensor grown under this condition has the best sensitivity. Therefore, a distance of 20 cm from the reaction zone is the condition for the manufacture of high performance sensors.

3.The shift of the absorption peak from 450 nm to 600 nm may be caused by different masses of reagents mixture varied ITO nanocolumns length, as shown in Fig. 4(a) - (d).

The shift of the absorption peak from 450 nm to 600 nm may be caused by different masses of reagents mixture varied the parameter of ITO nanocolumns (length, diameter, etc. ), as shown in Fig. 4(a) - (d).

4.Fig. 6.(d) - (f) show ITO-NM/U-bent/0.04/20, ITO-NM/U-bent-A/0.04/20, Gr/ITO-NM/U-bent-A/0.04/20 corresponding to the absorption spectrum of different refractive indices of alcohol, and the wavelength shift of these sensors in order was 11 nm, 15 nm and 21 nm, revealing that the annealing process enhanced the performance of the metamaterial sensor, and graphene also enhanced the performance of the metamaterial sensor.

 Fig. 6.(d) - (f) show ITO-NM/U-bent/0.04/20, ITO-NM/U-bent-A/0.04/20, Gr/ITO-NM/U-bent-A/0.04/20 corresponding to the absorption spectrum of different refractive indices of alcohol, and the wavelength shift of these sensors in order was 11 nm, 15 nm and 21 nm. We found that the performance of the sensors was constantly improving after the proposal methods. It explained that the annealing process did help to improve the sensitivity of the sensors due to the improvement of nano-array structure. And the deposition of graphene on the sensors also improved its performance mainly due to the relatively large specific surface area of graphene which could absorb more molecules.

5.From Fig. 9(a), we can clearly observe that lowest point of absorption spectrum for sensor+PBASE+probe DNA is 475 nm and there is almost no shift after adding the NC-DNA, suggesting that the obtained sensor has great performance in biological specificity.

From Fig. 9(a), we can clearly observe that lowest point of absorption spectrum for sensor+PBASE+probe DNA is 475 nm and there is almost no shift after adding the NC-DNA, which is nearly invariable and can be explained by the fact that there is a non-bonding reaction between the probe DNA and NC-DNA strands. Therefore, the refractive index change around the sensing region is relatively weak due to they cannot hybridize. Such phenomenon suggests that the obtained sensor has great performance in biological specificity.

6.When FC-DNA at different concentrations were added to the sensor which attached probe DNA, the absorption peak showed the absorption spectrum shifts to right.

When FC-DNA at different concentrations were added to the sensor which attached probe DNA, the absorption peak showed the absorption spectrum shifts to right. The shift of absorption spectrum can be explained by the electron-rich feature of DNA.

7.To further assess the performance of the sensor, the saturation and dynamics to biomolecules on metamaterial sensors were preliminarily explored. When the target DNA molecules were continuously captured by modified probe DNA, the RI around the sensing surface would increase. [42] Fig. 10(a) shows the variations of RI to 100 nM FC-DNA over time. The black curve presents the wavelength shift as a function of hybridization time in Fig. 10(b). Obviously, the sensor had an intense response to FC-DNA hybridization before 50 min, and then the response was gradually stabilized.

To further assess the performance of the sensor, the saturation and dynamics to biomolecules on metamaterial sensors were preliminarily explored. The responses of such sensors to FC-DNA with different concentration (1 nM - 100 nM) were detectwd and organized in real time. As shown in Fig. 10(a), the variations of absorption spectrum to 100 nM FC-DNA was taken as an example, which was shown in Fig. 10(b) (the black curve) that presented the wavelength shift as a function of hybridization time. When the target DNA molecules were continuously captured by modified probe DNA, the RI around the sensing surface would increase. [42] Obviously, the sensor had an intense response to FC-DNA hybridization before 50 min, and then the response was gradually stabilized. This indicated that the DNA hybridization reaction gradually progressed to saturation after a vigorous and rapid reaction.

2.The changes of alcohol concentration and other related details

1.Fig. 3(c) - (f) show the absorption spectra of different sensors by different concentration of alcohol solutions.

Fig. 3(c) - (f) show the absorption spectra of different sensors by different concentration of alcohol solutions ( refractive index (RI) from 1.3330 to 1.3634, respectively).

2.Figure 3. (a) The inner diameter of U-bent bare optical fiber. (b) AFM of Au film on the U-bent fiber. (c) - (f) The absorption spectra of ITO-NM/U-bent/0.04/12, ITO-NM/U-bent/0.04/16, ITO-NM/U-bent/0.04/20 and ITO-NM/U-bent/0.04/24 sensors .

Figure 3. (a) The inner diameter of U-bent bare optical fiber. (b) AFM of Au film on the U-bent fiber. (c) - (f) The absorption spectra of ITO-NM/U-bent/0.04/12, ITO-NM/U-bent/0.04/16, ITO-NM/U-bent/0.04/20 and ITO-NM/U-bent/0.04/24 sensors in alcohol solution with RI from 1.3330 to 1.3634.

3. Figure 4. (a) - (d) show the absorption spectra of ITO-NM/U-bent/0.01/20, ITO-NM/U-bent/0.04/20, ITO-NM/U-bent/0.07/20 and ITO-NM/U-bent/0.1/20 sensors, respectively.

4. Figure 4. (a) - (d) show the absorption spectra of ITO-NM/U-bent/0.01/20, ITO-NM/U-bent/0.04/20, ITO-NM/U-bent/0.07/20 and ITO-NM/U-bent/0.1/20 sensors in alcohol solution with RI from 1.3330 to 1.3634, respectively.

4.Figure 6. (a) Distribution of annealed Au particles. (b) Frontal profile of ITO (right figure shows the ITO with the pre-annealing process and the left exactly the opposite). (c) Surface morphology of Gr/ITO-NM/U-bent-A/0.04/20. (d) - (f) The absorption spectra of ITO-NM/U-bent/0.04/20, ITO-NM/U-bent-A/0.04/20, Gr/ITO-NM/U-bent-A/0.04/20 sensors.

Figure 6. (a) Distribution of annealed Au particles. (b) Frontal profile of ITO (right figure shows the ITO with the pre-annealing process and the left exactly the opposite). (c) Surface morphology of Gr/ITO-NM/U-bent-A/0.04/20. (d) - (f) The absorption spectra of ITO-NM/U-bent/0.04/20, ITO-NM/U-bent-A/0.04/20, Gr/ITO-NM/U-bent-A/0.04/20 sensors in alcohol solution with RI from 1.3330 to 1.3634.

Comment 5: Usually, ambient detection such as using alcohol tends to amplify the detection signature. In this context, the performance results reported as 690 nm/RIU is not particularly high. Neither is the data of DNA hybridization deemed properly analyzed. Authors need to discuss this in comparison with other studies (even outside fiberoptics based SPR). There are hundreds of related works, if not thousands.

   Fu, Hongyan, et al. "Graphene enhances the sensitivity of fiber-optic surface plasmon resonance biosensor." IEEE Sensors Journal 15.10 (2015): 5478-5482.

-       Ryu, Yeonsoo, et al. "Effect of coupled graphene oxide on the sensitivity of surface plasmon resonance detection." Applied optics 53.7 (2014): 1419-1426.

-       Grigorenko, A. N., Marco Polini, and K. S. Novoselov. "Graphene plasmonics." Nature photonics 6.11 (2012): 749.

-       Chung, Kyungwha, et al. "Systematic study on the sensitivity enhancement in graphene plasmonic sensors based on layer-by-layer self-assembled graphene oxide multilayers and their reduced analogues." ACS applied materials & interfaces 7.1 (2015): 144-151.

-       Wu, L., et al. "Highly sensitive graphene biosensors based on surface plasmon resonance." Optics express 18.14 (2010): 14395-14400.

Response:Thank you for your valuable and beneficial suggestion. And we attach great importance to your suggestion.

1.    The literature you mentioned describes the D-type sensor, the prism sensor. The evanescent field of U-type optical fibers is caused by the difference of internal and external refractive index. The formation of D-type and prism evanescent fields is different from that of U-type fibers. And their evanescent field is stronger than U-type optical fibers, which is due to the structure. But U-type optical fibers also have their own advantages. It is cheaper than D-type optical fibers and prisms, and its fabrication process is much simpler than D-type optical fibers and prisms. In addition, his detection is convenient and real-time, which is also the advantage of the structure. Therefore, although the evanescent field of U-type structure is not so prominent, we still choose U-type optical fiber as the experimental material.

2.    In terms of material structure, what we need to accomplish is a uniform nanoarray structure. But it is very difficult to complete and repeat the array structure. Of course, our degree of completion is not high, so the structure of the sensor is also one aspect, but this is also a bold innovation of the sensor structure. And we will continue to follow up the experiment.

The sensitivity of U-type optical fibers is much worse than that of D-type prisms. The following is the relevant literature.

(1) Zhang, Chao, et al. "U-bent fiber optic SPR sensor based on graphene/AgNPs." Sensors and Actuators B: Chemical 251 (2017): 127-133.

(2) Sai, V. V. R., Tapanendu Kundu, and Soumyo Mukherji. "Novel U-bent fiber optic probe for localized surface plasmon resonance based biosensor." Biosensors and Bioelectronics 24.9 (2009): 2804-2809.

(3) Liang, Gaoling, et al. "Plasma enhanced label-free immunoassay for alpha-fetoprotein based on a U-bend fiber-optic LSPR biosensor." Rsc Advances 5.31 (2015): 23990-23998.

(4) Wu, L., et al. "Highly sensitive graphene biosensors based on surface plasmon resonance." Optics express 18.14 (2010): 14395-14400.

(5) Ryu, Yeonsoo, et al. "Effect of coupled graphene oxide on the sensitivity of surface plasmon resonance detection." Applied optics 53.7 (2014): 1419-1426.

(6) Grigorenko, A. N., Marco Polini, and K. S. Novoselov. "Graphene plasmonics." Nature photonics 6.11 (2012): 749.

Reviewer 2 Report

The revised manuscript is an excellent and I have no serious criticisms reading methodology, result and interpretation result. The revised manuscript has a merit to published in Nanomaterials.
